# Beyond Transformations: Augmenting Anything for Image Super-Resolution via Diffusion Model

## Abstract

Image super-resolution (SR), aiming to restore accurate high-resolution images from low-resolution ones, plays a pivotal role in image processing. However, the performance of SR models is often hindered by conventional data augmentation and data degradation techniques. Conventional data augmentation methods for SR are typically limited to geometric transformations, lacking semantic richness. Traditional data degradation methods simulate degradation through a series of blurring, noise addition, compression, and resizing processes, lacking the complexity essential for robust model training. In this paper, based on pre-trained large-scale text-to-image diffusion models, we propose a novel data augmentation method and an innovative data degradation method in SR modeling. Our data augmentation method utilizes Stable Diffusion to modify image content at the semantic level for controlled data augmentation, enriching training datasets with nuanced variations while preserving the quality of the original images. Moreover, after fine-tuning Stable Diffusion with domain-matched data we further enhance the augmentation efficacy. Besides, by carefully designing control signals, our data degradation method utilizes diffusion to emulate degradation, simulating various unknown input corruptions to improve the performance of SR models across unfamiliar image degradation patterns. Our data augmentation method improves PSNR by 0.8 dB on the FFHQ dataset and by 0.28 dB on the Manga109 dataset for the SR tasks. Meanwhile, our data degradation technique has proven effective in significantly reducing artifacts in real-world SR imagery, distinctly exceeding the performance of traditional ones.

## 1 Introduction

Image super-resolution (SR), the task of reconstructing high-resolution images from their low-resolution counterparts, is pivotal in various fields, including medical imaging, satellite imagery, and video enhancement (Yang et al., 2007; Nasrollahi & Moeslund, 2014; Ledig et al., 2016). Data augmentation (DA) is crucial in SR modeling, particularly in scenarios with limited data, as it enhances dataset diversity, improves model generalization, and reduces overfitting. However, conventional geometric transformation-based DA techniques for the SR task (such as flipping and 90-degree rotation (Timofte et al., 2015)) often provide limited enhancement.

Intuitively, data augmentation is used to teach a model about invariances in the data domain (Cubuk et al., 2019), which helps shape a model's capacity to discern underlying patterns. Although traditional DA techniques for SR are effective at introducing geometric variability, they usually fail to provide rich semantic information and complex variations presented in real-world scenarios. The upper right portion of Figure 1 illustrates traditional DA methods' limitations in enhancing the model's image restoration performance in facial SR tasks. Though DA methods such as noise addition, color transformations, brightness/contrast adjustments, and more complex methods (Devries & Taylor, 2017; Yun et al., 2019; Zhang et al., 2017a; Hendrycks & Dietterich, 2019) have been widely proposed for high-level vision tasks, DA in low-level vision remains largely unexplored. Considering the importance of both local and global pixel relationships in low-level vision tasks (Yoo et al., 2020), applying DA strategies designed for high-level tasks directly to SR may degrade the quality of training data and negatively impact the efficacy of the SR models. For instance, the Jitter method

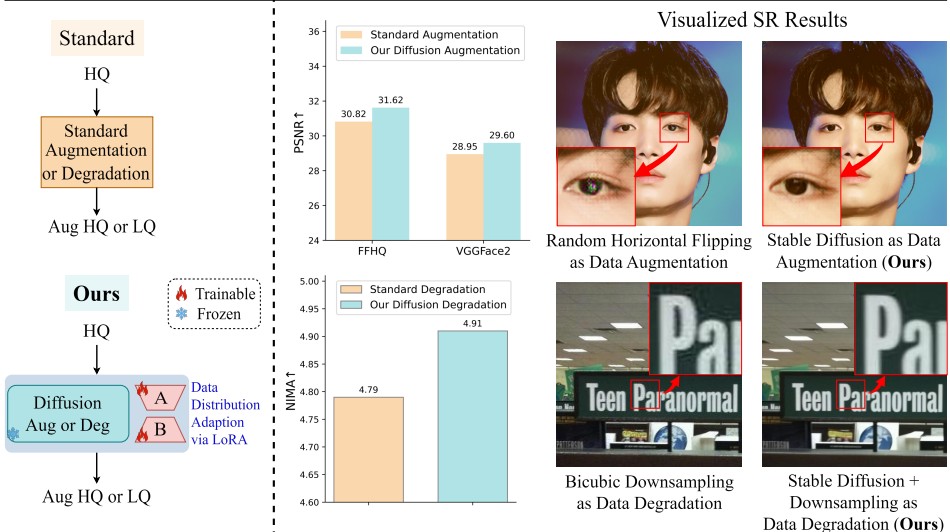

Figure 1: The left side highlights the main differences between our diffusion-based data augmentation and degradation methods and those of conventional approaches. The top row on the right side presents the facial image restoration results of the HiFaceGAN model (Yang et al., 2020b), comparing the use of traditional data augmentation methods with the use of Stable Diffusion. It can be observed that employing Stable Diffusion for data augmentation significantly enhances the fidelity of the SR outcomes. The bottom row on the right side demonstrates the restoration results of the SwinIR model (Liang et al., 2021) on real-world images, comparing the construction of HQ-LQ training data pairs using traditional degradation methods with those using Stable Diffusion. It is observed that employing Stable Diffusion for data degradation significantly reduces artifacts in the SR outcomes.

may disrupt the color space of images, violating the color patterns observed in the physical world, and the Image Erasing method (Zhong et al., 2017; Singh et al., 2018; Chen et al., 2020) could lead to the elimination of crucial information, adversely affecting the model's performance (Kumar et al., 2023).

Blind super-resolution (blind SR), aiming to super-resolve low-quality (LQ) images with unknown data degradation (DD) (Liu et al., 2021), contrasts with non-blind approaches that depend on explicit degradation information. Image degradation in the real world is often complex and not easily mimicked by direct mathematical models. This includes degradations caused by photographic equipment, such as camera blur, sensor noise, sharpening artifacts, and those resulting from the capture process, like motion blur. Furthermore, multiple sharing of the same image over networks can lead to cumulative quality loss. The lower right portion of Figure 1 illustrates that when the data degradation process of real-world images is ideally assumed to be bicubic downsampling, the SR model's restoration results in a higher incidence of artifacts.

Current blind SR methods can be broadly categorized into explicit and implicit modeling techniques (Wang et al., 2021), according to the ways of degradation modeling. Explicit modeling approaches (Zhang et al., 2017b; Gu et al., 2019; Michaeli & Irani, 2013; Bell-Kligler et al., 2019; Shocher et al., 2017; Cheng et al., 2020) rely on predefined degradation representations (blur, noise, JPEG compression, etc.), which, while straightforward, are frequently too idealized to hold true for the complex real-world degradation (Liu et al., 2021). Implicit modeling techniques, which utilize Generative Adversarial Networks (GANs) to simulate degradation processes by learning data distributions, require complex network designs and substantial computational resources (Yuan et al., 2018; Fritsche et al., 2019; Wei et al., 2020). While these methods are effective, their adaptability is constrained by the specific degradations present in the training data. Furthermore, the work of aligning high-quality (HQ) images with their LQ counterparts is laborious and time-consuming, resulting in a paucity of datasets that cover the breadth of real-world degradations.

Fortunately, recent breakthroughs in large-scale text-to-image models have introduced exciting new possibilities for image modification (Fan et al., 2023). As an influential model in this domain, Stable Diffusion (SD) has the remarkable ability to take an original image and, according to textual prompts and modification strength, apply various transformations to generate a new image.

In this work, we propose a novel data augmentation method and an innovative data degradation method in SR modeling. Based on Stable Diffusion models, our DA method effectively enriches the training datasets with diverse image variations, improving the generalizability of SR models on previously unseen data. Meanwhile, our DD method utilizes SD models to synthesize realistic HQ-LQ training pairs through the diffusion models' characteristic process of initial diffusion followed by denoising. It provides a fresh source of degradation knowledge for blind SR models that rely on learning from limited datasets to cope with unknown corruptions.

The primary contributions of this work are as follows:

- To the best of our knowledge, in the realm of super-resolution tasks, we are the first to propose the utilization of Stable Diffusion for data augmentation and degradation.
- We employ Stable Diffusion for semantic-level image content modification, achieving controlled data augmentation that introduces abundant variations to training datasets without compromising the original image quality, thereby boosting SR performance. Additionally, we fine-tune the SD model to align with the data distribution of specific domains, further enhancing SR performance through targeted data augmentation.
- We expand the use of Stable Diffusion to simulate controlled data degradation, thereby fortifying SR models against the variabilities and corruptions encountered in real-world imaging. This approach substantially minimizes restoration artifacts.

## 2 RELATED WORK

The pursuit of image SR has been a longstanding challenge in the field of computer vision. Traditional SR methods have transitioned from interpolation-based to learning-based approaches, with convolutional neural networks (CNNs) like SRCNN (Dong et al., 2014), EDSR (Lim et al., 2017), and SRGAN (Ledig et al., 2016) marking significant advances in quality through complex LR to HR mapping. Attention mechanisms, as introduced in the Transformer, further enhanced SR by focusing on multi-scale features (Chen et al., 2023; Zhang et al., 2023; Cao et al., 2021; Li et al., 2024; Yang et al., 2020a), achieving state-of-the-art reconstruction at the time. The Swin Transformer's hierarchical design and shift window mechanism have notably improved SR by capturing long-range dependencies, modeling both global and local contexts effectively (Liang et al., 2021; Conde et al., 2022; Choi et al., 2022), and setting new benchmarks in the field.

The core idea of data augmentation is to enhance the adequacy and diversity of training data through the creation of synthetic datasets (Yang et al., 2022), and incorporating potential invariances through DA is often more tractable than directly encoding them into the model architecture (Cubuk et al., 2019). Despite its importance, current DA methods for the SR task are primarily limited to geometric transformations, such as scaling, flipping and 90-degree rotation, which do not substantially contribute to semantic diversity in the dataset. This restricts the model's ability to learn complex mappings between LQ and HQ images, which is vital for accurate SR (Kumar et al., 2023; Russakovsky et al., 2014).

Beyond the constraints of traditional data augmentation methods, the understanding of how HQ images degrade to LQ images in most SR approaches is predicated on an ideal bicubic downsampling kernel, which deviates from actual degradation scenarios in the real world. Towards filling this gap, blind SR has garnered significant attention due to its ability to enhance image resolution without explicit knowledge of the degradation process. Blind SR techniques can be categorized into three primary classes: explicit modeling with external datasets, explicit modeling with single-image statistics, and implicit modeling through data distribution learning (Liu et al., 2021).

**External Dataset-based Explicit Modeling:** Methods like SRMD (Zhang et al., 2017b) and IKC (Gu et al., 2019) use diverse datasets in the training process to adapt to various blur and noise conditions. They perform well on trained degradations but struggle with novel ones. **Single-Image Statistic-based Explicit Modeling:** Approaches such as NPBSR (Michaeli & Irani, 2013), Kernel-

GAN (Bell-Kligler et al., 2019), ZSSR (Shocher et al., 2017), and DGDML-SR (Cheng et al., 2020) exploit image internal statistics for kernel estimation and SR without external data. They rely on the presence of recurring image patches, which may be scarce in diverse or monotonous images. **Implicit Modeling via Data Distribution Learning:** CinCGAN (Yuan et al., 2018), FSSR (Fritsche et al., 2019), and DASR (Wei et al., 2020) use GANs to implicitly learn degradation models from external datasets. They generate LR images with realistic degradations for SR training but can produce artifacts unsuitable for real-world use.

## 3 PROPOSED METHOD

In the realm of super-resolution, such a low-level vision task, we are the first to propose leveraging the content generation capability of Stable Diffusion to implement both data augmentation and data degradation processes. The proposed data augmentation method depicted in Figure 2A is utilized to enrich the training dataset, thereby enhancing the SR model's image restoration capability in unseen scenarios. Meanwhile, our data degradation method depicted in Figure 2B is employed to diversify the degradation forms in the HQ-LQ image pairs of training data, thus improving the SR model's performance on tasks with unknown degradation types.

### 3.1 CONTROLLED DATA AUGMENTATION METHOD

In contrast to high-level vision tasks, SR places a higher demand on the quality, particularly the resolution, of images in the training set. Higher-quality training images contain more visually pleasing texture details, which contribute to better model training outcomes. Therefore, to preserve the quality of images, data augmentation for SR typically introduces only geometric transformations. However, such methods are insufficient to augment the information contained in images, thereby inadequately enhancing the richness of the dataset. More complex augmentation methods, like noise addition, color transformations, and brightness/contrast adjustments, may disrupt the local and global relationships among pixels, thus they are not suitable for the SR task. Therefore, there is a need for a data augmentation method that preserves image quality, effectively increases image information, and ideally is convenient to operate.

Diffusion models learn the underlying data distribution through successive iterations of forward diffusion and reverse denoising. This process enables the efficient generation of a diverse set of samples, closely aligning with the target data distribution. Stable Diffusion serves as a large-scale pre-trained text-to-image diffusion model that encapsulates extensive image prior information. Our data augmentation method aims to infuse image prior information inherent in Stable Diffusion into the original images during the modification process, thereby enriching the information content of the training data.

Figure 2A illustrates our DA workflow. Stable Diffusion takes an original $H \times W$ image from the training dataset and, guided by control signals such as textual prompts, negative prompts, and modification strength, encodes the image into a noisy latent space. Then it predicts and removes this noise based on the provided control signals, producing an enhanced latent representation. A decoder subsequently reconstructs this representation into an augmented image of the same $H \times W$ dimensions, introducing content variations while preserving image clarity.

As depicted in Figure 2A, with the textual prompt set to "yellow ducks, with high resolution," negative prompts including "blurry, noisy, deformed, poor details, distorted, flat, jarring, pixelated," and a modification strength of 0.6, the original image is transformed into three distinct images. Among these modifications, the kumquats in the original image are converted into some yellow ducks to varying extents, achieving a unique effect unattainable by conventional data augmentation methods. Following DA, the modified images are downsampled to create their LQ counterparts. Subsequently, random cropping is applied to the HQ-LQ image pairs within the same region, yielding image pairs with appropriate size to train SR models. Prior to these steps, fine-tuning the Stable Diffusion according to the distribution of input images' domain and employing it for data augmentation further enhances the SR outcomes.

It is worth noting that our data augmentation process is independent of traditional ones. Applying conventional DA techniques to images either before or after utilizing our method is entirely feasible and may yield enhanced augmentation outcomes.

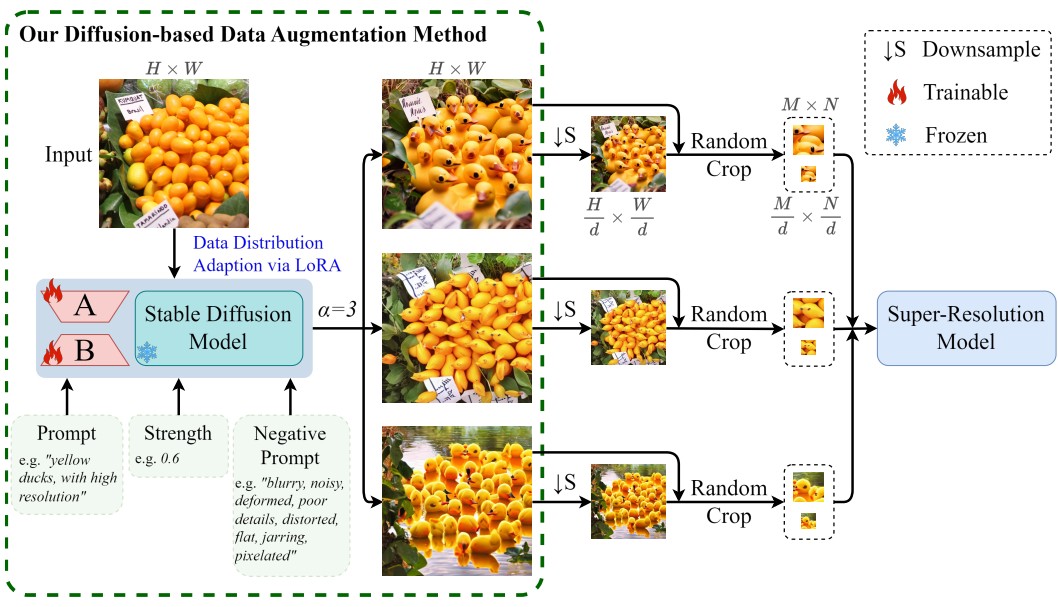

(A) The proposed Data Augmentation method

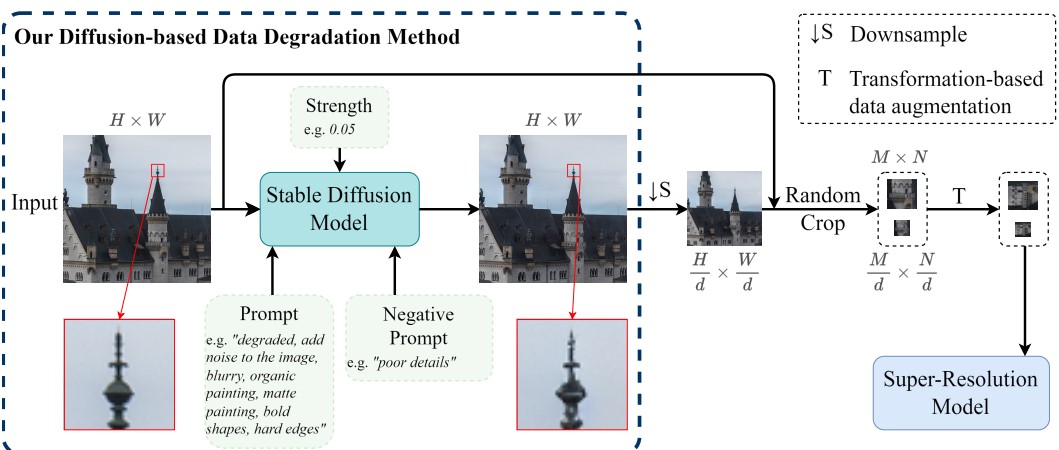

(B) The proposed Data Degradation method

Figure 2: Demonstration of the proposed data augmentation and data degradation method in SR modeling. As shown in subfigure (A), our data augmentation method leverages the generative capabilities of Stable Diffusion to modify original images, yielding $\alpha$ (the expansion factor) augmented outcomes, by appropriately setting prompts, negative prompts, and modification strength. Then, downsample the augmented results to obtain corresponding low-quality (LQ) images. Each pair of high-quality (HQ) and LQ images, after random cropping, is utilized to train the SR model. Fine-tuning the Stable Diffusion on data from the same domain as the input images yields better augmentation outcomes. Meanwhile, as shown in subfigure (B), our data degradation method utilizes Stable Diffusion to directly effectuate degradation on the original images by appropriately setting prompts, negative prompts, and strength. Subsequently, after the requisite downsampling process, the original images and their downsampled counterparts form HQ-LQ image pairs. After random cropping and transformation-based data augmentation, which includes random flipping and random 90-degree rotation, these pairs are sent to train the SR model.

## 3.2 Controlled Data Degradation Module

Degradation information, as knowledge embedded in the training data, plays a crucial guiding role in the process of recovering HQ images from LQ ones. Existing blind SR works either model real-world image degradations as straightforward blends of blurring, noise, and JPEG compression, or utilize GANs to learn more implicit degradation patterns from limited datasets. The former approach is often overly idealized, failing to capture complex degradation processes, while the latter tends to be computationally expensive and limited to the degradations present within training datasets, with poor generalization to out-of-distribution images. It is worth noting that, to the best of our knowledge, training datasets containing precisely paired HQ-LQ images generated by real-world degradation processes are exceedingly rare. Therefore, we require data degradation methods that more effectively simulate the complex degradation processes in real-world scenarios.

We observed that when the modification strength is set to a low value (such as 0.05), the alterations made by Stable Diffusion to the original image manifest as random, yet subtle, distortions and blurring of textural details, without significant changes to the overall semantic information and color space of the image. The local and global relationships between pixels remain essentially unchanged. This observation has inspired us to explore whether the image modification process of Stable Diffusion could serve as a form of data degradation. Figure 2B illustrates our methodology for inducing degradation in HQ images. Given an image of dimensions $H \times W$ from the dataset, the SD model is guided by specific prompts, negative prompts, and a carefully calibrated modification strength—our experience indicates that a lower modification strength tends to yield superior outcomes.

As shown in Figure 2B, with the prompt set to "degraded, add noise to the image, blurry, organic painting, matte painting, bold shapes, hard edges," a negative prompt of "poor details," and a modification strength of 0.05, subtle changes are introduced to the textural details of the lightning rod on the roof, while the overall image maintains a high degree of consistency with the original one. The outputs of Stable Diffusion, after an essential downsampling process, serve as the LQ images, with the original images acting as the HQ ones. The HQ-LQ image pairs, after random cropping and transformation-based data augmentation, ultimately serve as training data for SR models.

It is noteworthy that, just like our proposed data augmentation method, our data degradation process is also independent of traditional DD processes. Applying conventional DD techniques to images either before or after utilizing our method is entirely feasible and can effectively address more complex real-world degradation scenarios.

## 4 Experiments

In this section, we delineate the training datasets and corresponding parameter configurations of Stable Diffusion tailored for various downstream scenarios. These scenarios encompass SR tasks for facial images, anime images, and blind SR tasks for real-world images. Moreover, a set of ablation experiments, focusing on the expansion factor, whether the SD model has been fine-tuned, and the modification strength, demonstrate the individual impact of these factors on data augmentation effects.

### 4.1 Super-Resolution Task for Facial Images

When exploring the efficacy of our proposed data augmentation method for facial SR tasks, we select the HiFaceGAN and ESRGAN (Wang et al., 2018) as base models for the reconstruction of HQ facial images. Traditional DA methods typically include only random horizontal flipping, without additional forms of augmentation, which is attributed to the distinctive structure of the human face. We design comprehensive comparative experiments for analysis. The original training data consists of 10,000 images from the FFHQ (Karras et al., 2018) dataset, while the testing data includes an additional 1,000 images from the FFHQ dataset and 2,000 images from the VGGFace2 (Cao et al., 2017) dataset. The DA conditions for each experimental group are as follows:

(1) As a fundamental control group, only random horizontal flipping is utilized to augment the original images in the training set, with each image corresponding to a single augmented result. The augmented training data still consists of 10,000 images. For these results, we utilize 4x bicubic

Table 1: Results on FFHQ and VGGFace2.

| Base Model | Data Augmentation | Test Set/Data Degradation | PSNR↑ | SSIM↑ | FID↓ | LPIPS↓ | DISTS↓ | NIQE↓ |
|---|---|---|---|---|---|---|---|---|
| HiFaceGAN | Transformation-based | FFHQ/4x | 30.82 | 0.8526 | **10.24** | **0.0828** | 0.0830 | **3.67** |
| | | FFHQ/4-8x | 28.86 | 0.7971 | **41.78** | **0.2099** | **0.1589** | **5.48** |
| | | VGGFace2/4x | 28.95 | 0.8164 | 40.00 | **0.1271** | 0.3558 | **4.75** |
| | **Diffusion-based (Ours)** | FFHQ/4x | **31.62** | **0.8640** | 11.4 | 0.0889 | **0.0803** | 4.09 |
| | | FFHQ/4-8x | **29.05** | **0.8029** | 47.16 | 0.2242 | 0.1613 | 6.21 |
| | | VGGFace2/4x | **29.60** | **0.8292** | 21.32 | 0.1341 | **0.3557** | 5.11 |
| ESRGAN | Transformation-based | FFHQ/4x | 29.32 | 0.8148 | **8.59** | **0.0811** | **0.0716** | **3.47** |
| | | VGGFace2/4x | 25.28 | 0.6926 | 70.50 | **0.2812** | 0.2178 | **5.04** |
| | **Diffusion-based (Ours)** | FFHQ/4x | **29.63** | **0.8196** | 9.92 | 0.0886 | 0.0797 | 3.57 |
| | | VGGFace2/4x | **25.66** | **0.7065** | **67.17** | 0.2834 | **0.2122** | 5.35 |

downsampling to construct its corresponding LQ image. The SR model's total number of epochs was set to 50, with all 10,000 augmented image pairs being fed into the model in each epoch.

(2) We fine-tuned the SD model using 30,000 facial images from the CelebAMask-HQ dataset (Lee et al., 2019), aiming to enhance its ability to generate more realistic facial textures and details. For data augmentation, each original image in the training set was processed by the fine-tuned SD model with the prompt set to "a picture of natural and detailed human face with high resolution," and the negative prompt as "blurry, noisy, deformed, poor details, distorted, flat, jarring, pixelated." The modification strength was randomly set between 0 and 0.55, with each original image corresponding to ten augmented results. The augmented training data expanded to 100,000 images. For each image in the set, we utilize 4x bicubic downsampling to construct its corresponding LQ image. The SR model's total epoch count was set to 5, feeding all 100,000 augmented training images into the model in each epoch, thereby ensuring that the total number of iterations was equivalent to that in group (1).

Further details on the experimental settings and results of additional groups are presented in the ablation study section.

As demonstrated in Table 1, our DA method outperforms traditional approaches in terms of PSNR and SSIM, leading to significant improvements in SR performance.

## 4.2 SUPER-RESOLUTION TASK FOR ANIME IMAGES

In exploring the effectiveness of our proposed data augmentation method for anime image SR tasks, we select the SwinIR and HAT (Chen et al., 2022) as base models for the reconstruction of HQ anime images. Traditional data augmentation techniques include random horizontal flipping, random vertical flipping, and random 90-degree rotations.

We design sufficient comparative experiments for analysis. The original training data consists of 5,800 images from the animeSR dataset (Ye, 2021), while the testing data includes another 650 images from the animeSR dataset, 109 images from the Manga109 (Matsui et al., 2015) dataset and 1,000 images from the iCartoonFace (Zheng et al., 2019) dataset. The DA conditions for each experimental group are as follows:

(1) Serving as a fundamental control group, a combination of random horizontal flipping, random vertical flipping, and random 90-degree rotation is utilized to augment the original images in the training set, with each image corresponding to a single augmented result. The augmented training data is maintained at 5,800 images. The total number of iterations for the SR model's training is set to 500,000, repeatedly learning from these 5,800 images.

(2) The Stable Diffusion model is employed for data augmentation of the original images. When each image is processed by Stable Diffusion, the prompt is set to "an image of Cartoon, with high resolution," and the negative prompt is "blurry, noisy, deformed, poor details, distorted, flat, jarring, pixelated." The modification strength is set to a random number between 0 and 0.3, with each original image corresponding to ten augmented results. The augmented training data expands to include 58,000 images. The total number of iterations for the SR model's training is set to 500,000 as well.

Table 2: Results on animeSR, Manga109 and iCartoonFace.

| Base Model | Scale | Data Augmentation | Test Set | PSNR↑ | SSIM↑ | LPIPS↓ | DISTS↓ | NIMA↑ |
|---|---|---|---|---|---|---|---|---|
| SwinIR | 2x | Transformation-based | animeSR | 32.35 | **0.9395** | 0.0722 | 0.1013 | **4.87** |
| | | | Manga109 | 31.02 | 0.9351 | **0.0753** | 0.0922 | 5.09 |
| | | | iCartoonFace | 33.31 | 0.9494 | **0.0556** | 0.1284 | **4.18** |
| | | **Diffusion-based (Ours)** | animeSR | **32.49** | 0.9389 | 0.0857 | 0.1060 | 4.62 |
| | | | Manga109 | **31.30** | **0.9443** | 0.0820 | **0.0794** | **5.16** |
| | | | iCartoonFace | **34.21** | **0.9537** | 0.0581 | 0.1115 | **4.18** |
| HAT | 4x | Transformation-based | animeSR | 28.32 | 0.8649 | **0.1532** | 0.1718 | 4.99 |
| | | | Manga109 | **24.84** | 0.8501 | **0.1591** | 0.1414 | 5.32 |
| | | | iCartoonFace | 29.52 | 0.8982 | **0.1148** | 0.1690 | 4.43 |
| | | **Diffusion-based (Ours)** | animeSR | **28.55** | **0.8688** | 0.1660 | **0.1718** | 4.78 |
| | | | Manga109 | 24.75 | **0.8546** | 0.1683 | 0.1426 | **5.33** |
| | | | iCartoonFace | **30.00** | **0.9022** | 0.1149 | **0.1589** | 4.27 |

Further details on the experimental settings and results of additional groups are presented in the ablation study section.

As demonstrated in Table 2, our DA method outperforms traditional approaches in terms of PSNR, SSIM, and some other metrics, leading to improvements in SR performance to some extent.

## 4.3 BLIND SUPER-RESOLUTION TASK FOR REAL-WORLD IMAGES

When exploring the efficacy of our proposed data degradation method for blind SR tasks in real-world scenarios, we select the SwinIR and MambaIR (Guo et al., 2024) as base models for the reconstruction of HQ images. Traditional DD methods include bicubic downsampling, noise addition, and blurring, among others.

We design a series of comparative experiments: the HQ images are sourced from the cropped DIV2K dataset, totaling 27,000 images, while the testing data includes 3,000 images from the ADE20K dataset (Zhou et al., 2016). The data degradation conditions for each group are as follows:

(1) As a fundamental control group, we utilize solely downsampling to reduce the size of HQ images, resulting in 27,000 pairs of HQ-LQ images.

(2) We initially employ Stable Diffusion for data degradation of HQ images with prompts set to "degraded, add noise to the image, blurry, organic painting, matte painting, bold shapes, hard edges," and a negative prompt of "poor details," with modification strength randomly set between 0 and 0.1. Subsequently, the outputs of SD are further downsampled to reduce size.

(3) As an enhanced control group, we first apply 4x bicubic downsampling to degrade HQ images, followed by downsampling to reduce size.

(4) We begin with Stable Diffusion with the same settings as in (2). Then, we apply 4x bicubic downsampling to the outputs of SD, concluding with downsampling to reduce size.

(5) As another enhanced control group, we first add random noise to degrade the HQ images, followed by downsampling to reduce size.

(6) We first employ Stable Diffusion with the same settings as in (2). Then, we add random noise to the outputs of SD, followed by downsampling to reduce size.

In practical scenarios, when we deal with images from real-world scenes, they are already the results of unknown degradation processes, so no one truly knows what their corresponding HQ ground truths are. Therefore, in our experiments, we treat all images from the testing dataset as LQ images obtained through unknown degradation processes and use various SR models to restore them. The visual quality of the SR results is the most critical criterion for evaluating the effectiveness of the restoration. Figures 3A and 3B respectively present the restoration results from SwinIR and MambaIR on ADE20K. The bottom rows on the right side of the two figures employ our proposed method. It can be observed that incorporating Stable Diffusion for data degradation significantly reduces artifacts in the restoration results compared to their standard counterparts in the rows above.

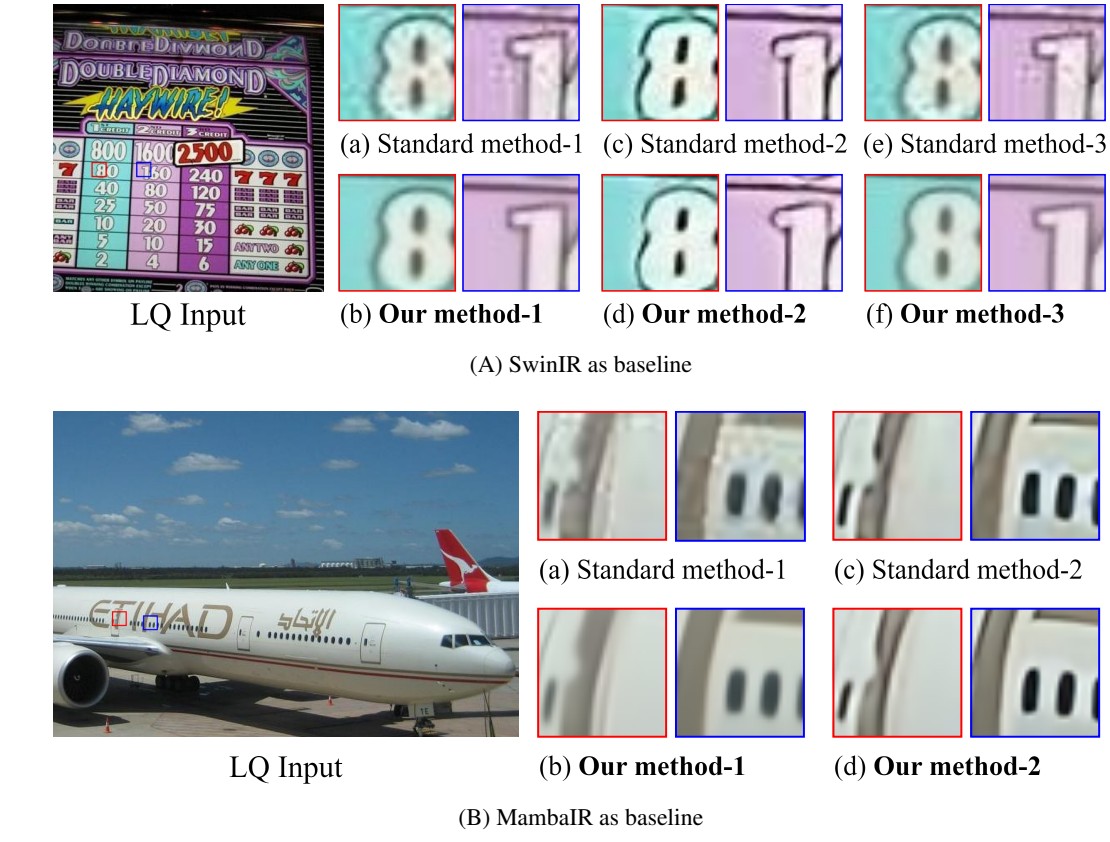

(a) Standard method-1   (c) Standard method-2   (e) Standard method-3

LQ Input   (b) **Our method-1**   (d) **Our method-2**   (f) **Our method-3**

(A) SwinIR as baseline

(a) Standard method-1   (c) Standard method-2

LQ Input   (b) **Our method-1**   (d) **Our method-2**

(B) MambaIR as baseline

Figure 3: A visual demonstration of the SwinIR and MambaIR models' performance in restoring images with unknown degradation from real-world scenarios, trained using HQ-LQ pairs constructed from various data degradation methods. (a) resizing (downsampling), (b) first employing Stable Diffusion then resizing, (c) first using 4x bicubic downsampling then resizing, (d) first employing Stable Diffusion then using 4x bicubic downsampling and resizing, (e) first adding noise then resizing, (f) first employing Stable Diffusion then adding noise and resizing.

## 4.4 ABLATION INVESTIGATION

We conduct ablation studies to investigate the impact of three key factors on SR outcomes: the expansion factor, the fine-tuning of Stable Diffusion, and the modification strength.

### 4.4.1 THE EXPANSION FACTOR

In the domain of SR, such a low-level vision task, traditional data augmentation methods based on geometric transformations not only struggle to enrich the information within the original images but also yield a limited number of augmentation outcomes. For instance, when the augmentation method involves a combination of random horizontal and vertical flips, a single original image can be expanded to at most four results. In contrast, Stable Diffusion, due to the randomness inherent in its diffusion and denoising processes, can expand a single original image into an infinite number of results. Table 3 shows the anime SR results from SwinIR, indicating that a higher expansion factor typically yields better results. The BOLD and UNDERLINE in the table indicates the best and second best results respectively.

### 4.4.2 FINE-TUNED OR NOT FINE-TUNED

Apart from the textual prompts, negative prompts, and modification strength settings, the inherent characteristics of Stable Diffusion itself can also influence the final SR outcomes. These characteristics are primarily determined by the specific network architecture of SD and the data used for its

Table 3: Different expansion factor.

| Data Augmentation | Expansion Factor | Test Set/ scale | PSNR↑ | SSIM↑ |
|---|---|---|---|---|
| Transformation | 1 | animeSR/2x | 32.35 | 0.9395 |
|  |  | Manga109/2x | 31.02 | 0.9351 |
|  |  | iCartoonFace/4x | 28.65 | 0.8844 |
| **Diffusion (Ours)** | 1 | animeSR/2x | 32.49 | 0.9397 |
|  |  | Manga109/2x | 31.04 | 0.9429 |
|  |  | iCartoonFace/4x | 28.76 | 0.8816 |
| **Diffusion (Ours)** | 10 | animeSR/2x | 32.49 | 0.9389 |
|  |  | Manga109/2x | 31.30 | 0.9443 |
|  |  | iCartoonFace/4x | 29.40 | 0.8916 |

Table 4: Fine-tuned or not

| Data Augmentation | Fine-tune | Test Set/ Data Degradation | PSNR↑ | SSIM↑ |
|---|---|---|---|---|
| Transformation | - | FFHQ/4x | 30.82 | 0.8526 |
|  |  | FFHQ/4-8x | 28.86 | 0.7971 |
|  |  | VGGFace2/4x | 28.95 | 0.8164 |
| **Diffusion (Ours)** | w/o | FFHQ/4x | **31.28** | 0.8574 |
|  |  | FFHQ/4-8x | 29.01 | 0.8007 |
|  |  | VGGFace2/4x | 29.23 | 0.8201 |
| **Diffusion (Ours)** | w/ | FFHQ/4x | 31.24 | **0.8575** |
|  |  | FFHQ/4-8x | **29.04** | **0.8022** |
|  |  | VGGFace2/4x | **29.28** | **0.8223** |

Table 5: Impact of different Modification Strength on facial SR task

| Data Augmentation | Strength | Test Set/Data Degradation | PSNR↑ | SSIM↑ | FID↓ | LPIPS↓ | DISTS↓ | NIQE↓ |
|---|---|---|---|---|---|---|---|---|
| Transformation-based | - | FFHQ/4x | 30.82 | 0.8526 | **10.24** | **0.0828** | **0.0830** | **3.67** |
|  |  | FFHQ/4-8x | 28.86 | 0.7971 | 41.78 | **0.2099** | **0.1589** | 5.48 |
|  |  | VGGFace2/4x | 28.95 | 0.8164 | 40.00 | **0.1271** | 0.3558 | **4.75** |
| **Diffusion-based (Ours)** | 0-0.3 | FFHQ/4x | **31.10** | **0.8591** | 14.37 | 0.0995 | 0.0959 | 4.10 |
|  |  | FFHQ/4-8x | 28.86 | 0.7994 | 44.51 | 0.2200 | 0.1651 | 5.67 |
|  |  | VGGFace2/4x | 29.17 | 0.8211 | 31.70 | 0.1291 | 0.1324 | 5.05 |
| **Diffusion-based (Ours)** | 0.3-0.6 | FFHQ/4x | 31.01 | 0.8552 | **10.24** | 0.0840 | 0.0850 | 3.83 |
|  |  | FFHQ/4-8x | **28.93** | **0.8003** | 41.81 | 0.2155 | 0.1614 | 5.72 |
|  |  | VGGFace2/4x | **29.24** | **0.8222** | 27.41 | 0.1276 | 0.1275 | 4.89 |
| **Diffusion-based (Ours)** | 0.6-1 | FFHQ/4x | 30.96 | 0.8569 | 12.42 | 0.0940 | 0.0937 | 3.90 |
|  |  | FFHQ/4-8x | 28.85 | 0.7980 | 43.87 | 0.2170 | 0.1642 | **5.05** |
|  |  | VGGFace2/4x | 29.15 | 0.8212 | **25.07** | 0.1290 | 0.1300 | 4.86 |

pre-training. In this work, we take a data-driven approach and fine-tune the SD model using 30,000 facial images from CelebAMask-HQ. The results from HifaceGAN shown in Table 4 indicate that using fine-tuned SD for data augmentation typically further enhances the performance of SR models.

### 4.4.3 THE MODIFICATION STRENGTH

In the application of Stable Diffusion for data augmentation, the modification strength is an important parameter that significantly influences the resulting images. An increased strength value bestows greater "creativity" on the model, yielding images that diverge from the original; a value of 1.0 implies near-total disregard for the initial image. Conversely, a reduced strength value generates images that closely resemble the original. Table 5 shows the impact of different modification strengths on the facial SR results when using HiFaceGAN. It can be observed that a modification strength range of 0.3 to 0.6 may be more suitable for the facial SR task.

## 5 CONCLUSION

Our exploration of the Stable Diffusion models in super-resolution tasks has yielded promising results, highlighting their potential in data augmentation and data degradation. The novel approach of utilizing Stable Diffusion for data augmentation and degradation has significantly enriched the content and degradation information within the training datasets, thereby achieving superior generalization and restoration quality. This work not only advanced the state-of-the-art in image SR but also laid the groundwork for future research on Stable Diffusion in other low-level vision tasks.

A plethora of experimental results has led us to believe that refining strategies for controlling signals such as textual prompts, along with more advanced generative models, will yield further exciting benefits. As we delve deeper into the field of imaging science, the role of large-scale pre-trained text-to-image models becomes increasingly crucial. Our findings set a new precedent in the field, advocating for the integration of powerful generative techniques to craft robust visual algorithms capable of meeting the complexities of contemporary imaging demands.

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

# A    APPENDIX

In the appendix, we first present additional experimental results to complement the content of the main text. These include SR results for anime images (A.1), experiments exploring the impact of the expansion factor during data augmentation on facial SR task (A.2), and more blind SR outcomes for real-world images (A.3). The remainder of this section (A.4) details the implementation of our experiments.

## A.1    RESULTS OF ANIME SR TASK

Table 6 presents a more detailed analysis of the anime SR experiment results, demonstrating our data augmentation method's significant advantage in the more challenging 4x SR task.

Table 6: Anime SR results on animeSR, Manga109 and iCartoonFace.

| Base Model | Data Augmentation | Scale | Test set | PSNR↑ | SSIM↑ | LPIPS↓ | MUSIQ↑ | DISTS↓ | NIMA↑ |
|---|---|---|---|---|---|---|---|---|---|
| SwinIR | Transformation-based | 2x | animeSR | 32.35 | **0.9395** | **0.0722** | **59.40** | **0.1013** | **4.87** |
| | | | Manga109 | 31.02 | 0.9351 | **0.0753** | **72.89** | 0.0922 | 5.09 |
| | | | iCartoonFace | 33.31 | 0.9494 | **0.0556** | **51.27** | 0.1284 | **4.18** |
| | | 4x | animeSR | 27.67 | 0.8523 | **0.1892** | **47.82** | 0.1857 | **4.59** |
| | | | Manga109 | 25.25 | 0.8477 | **0.1677** | **67.66** | **0.1341** | 5.37 |
| | | | iCartoonFace | 28.65 | 0.8844 | 0.1343 | **45.19** | 0.1617 | **4.18** |
| | **Diffusion-based (Ours)** | 2x | animeSR | **32.49** | 0.9389 | 0.0857 | 55.64 | 0.1060 | 4.62 |
| | | | Manga109 | **31.30** | **0.9443** | 0.0820 | 70.85 | **0.0794** | **5.16** |
| | | | iCartoonFace | **34.21** | **0.9537** | 0.0581 | 47.76 | **0.1115** | **4.18** |
| | | 4x | animeSR | **28.07** | **0.8573** | 0.1984 | 46.43 | **0.1834** | 4.46 |
| | | | Manga109 | **25.50** | **0.8525** | 0.1738 | 66.22 | 0.1368 | **5.37** |
| | | | iCartoonFace | **29.40** | **0.8916** | **0.1319** | 43.02 | **0.1536** | 4.10 |
| HAT | Transformation-based | 2x | animeSR | **34.36** | **0.9524** | **0.0617** | **61.52** | 0.1082 | **5.09** |
| | | | Manga109 | 30.97 | **0.9479** | **0.0708** | **72.10** | 0.0927 | 5.10 |
| | | | iCartoonFace | 35.06 | **0.9592** | 0.0492 | 50.69 | 0.1224 | **4.32** |
| | | 4x | animeSR | 28.32 | 0.8649 | **0.1532** | **57.11** | **0.1718** | **4.99** |
| | | | Manga109 | **24.84** | 0.8501 | **0.1591** | **71.68** | **0.1414** | 5.32 |
| | | | iCartoonFace | 29.52 | 0.8982 | **0.1148** | **51.17** | 0.1690 | **4.43** |
| | **Diffusion-based (Ours)** | 2x | animeSR | 33.46 | 0.9470 | 0.0726 | 58.57 | **0.1038** | 4.85 |
| | | | Manga109 | **31.37** | 0.9478 | 0.0718 | 71.95 | **0.0855** | **5.13** |
| | | | iCartoonFace | **35.20** | 0.9589 | **0.0484** | 48.42 | **0.1065** | 4.20 |
| | | 4x | animeSR | **28.55** | **0.8688** | 0.1660 | 53.02 | **0.1718** | 4.78 |
| | | | Manga109 | 24.75 | **0.8546** | 0.1683 | 69.63 | 0.1426 | **5.33** |
| | | | iCartoonFace | **30.00** | **0.9022** | 0.1149 | 48.19 | **0.1589** | 4.27 |

## A.2    IMPACT OF THE EXPANSION FACTOR ON FACIAL SR TASK

Table 7 illustrates the impact of the expansion factor during data augmentation on facial SR outcomes, revealing that a higher expansion factor generally leads to greater improvements in SR performance. The BOLD and UNDERLINE in the table indicates the best and second best results respectively. Stable Diffusion can augment an original image by arbitrary multiples, an advantage not present in traditional data augmentation methods.

## A.3    BLIND SR RESULTS FOR REAL-WORLD IMAGES

Figures 4 and 5 respectively demonstrate the impact of different HQ-LQ training data pair constructions on the image restoration performance of models when using SwinIR and MambaIR as base models. It is observed that incorporating the image modification operations of Stable Diffusion during the degradation process from HQ to LQ images can cover more unknown degradations present in real-world scenarios, thereby enhancing the SR models' restorative performance and significantly reducing artifacts in the restoration outcomes.

Table 7: Facial SR results on FFHQ and VGGFace2.

| Base Model | Data Augmentation | Expansion Factor | Test set Data Degradation | PSNR↑ | SSIM↑ | FID↓ | LPIPS↓ | DISTS↓ | NIQE↓ |
|---|---|---|---|---|---|---|---|---|---|
| HiFaceGAN | Transformation-based | 1 | FFHQ/4x | 30.82 | 0.8526 | **10.24** | **0.0828** | 0.0830 | **3.67** |
| | | | FFHQ/4-8x | 28.86 | 0.7971 | **41.78** | **0.2099** | **0.1589** | **5.48** |
| | | | VGGFace2/4x | 28.95 | 0.8164 | 40.00 | **0.1271** | 0.3558 | **4.75** |
| | **Diffusion-based (Ours)** | 1 | FFHQ/4x | 31.24 | 0.8575 | 13.47 | 0.1008 | 0.0945 | 3.94 |
| | | | FFHQ/4-8x | 29.04 | 0.8022 | 48.70 | 0.2276 | 0.1675 | 5.82 |
| | | | VGGFace2/4x | 29.28 | 0.8223 | 25.40 | 0.1392 | 0.3569 | 4.88 |
| | **Diffusion-based (Ours)** | 10 | FFHQ/4x | **31.62** | **0.8640** | **11.40** | 0.0889 | **0.0803** | 4.09 |
| | | | FFHQ/4-8x | **29.05** | **0.8029** | 47.16 | 0.2242 | 0.1613 | 6.21 |
| | | | VGGFace2/4x | **29.60** | **0.8292** | 21.32 | 0.1341 | **0.3557** | 5.11 |
| ESRGAN | Transformation-based | 1 | FFHQ/4x | 29.32 | 0.8148 | **8.59** | **0.0811** | **0.0716** | **3.47** |
| | | | VGGFace2/4x | 25.28 | 0.6926 | 70.50 | **0.2812** | 0.2178 | **5.04** |
| | **Diffusion-based (Ours)** | 1 | FFHQ/4x | 29.22 | 0.8093 | 11.12 | 0.1026 | 0.0870 | 3.52 |
| | | | VGGFace2/4x | 25.55 | 0.6994 | **66.13** | 0.2883 | 0.2133 | 5.40 |
| | **Diffusion-based (Ours)** | 10 | FFHQ/4x | **29.63** | **0.8196** | 9.92 | 0.0886 | 0.0797 | 3.57 |
| | | | VGGFace2/4x | 25.66 | **0.7065** | 67.17 | 0.2834 | **0.2122** | 5.35 |

## A.4 EXPERIMENTAL IMPLEMENTATION

### A.4.1 DIFFUSION-BASED DATA AUGMENTATION AND DEGRADATION

In this work, we employed Stable Diffusion for controlled data augmentation and degradation. We have utilized "CompVis/stable-diffusion-v1-4" (available at `https://huggingface.co/CompVis/stable-diffusion-v1-4`) and "runwayml/stable-diffusion-v1-5". Although the model parameters for "runwayml/stable-diffusion-v1-5" are currently inaccessible due to certain reasons, updated versions of Stable Diffusion continue to be trained and released (can be found on website `https://huggingface.co/`). We believe that with the ongoing updates and advancements of Stable Diffusion, its advantages in data augmentation will become even more pronounced.

### A.4.2 SUPER-RESOLUTION

In this work, we utilized the HiFaceGAN, ESRGAN, SwinIR, HAT, and MambaIR as base models for various super-resolution tasks.

**HiFaceGAN** is available at `https://github.com/Lotayou/Face-Renovation` and `https://github.com/XPixelGroup/BasicSR`.

**ESRGAN** is available at `https://github.com/XPixelGroup/BasicSR`.

**SwinIR** is available at `https://github.com/JingyunLiang/SwinIR` and `https://github.com/XPixelGroup/BasicSR`.

**HAT** is available at `https://github.com/XPixelGroup/HAT`.

**MambaIR** is available at `https://github.com/csguoh/MambaIR`.

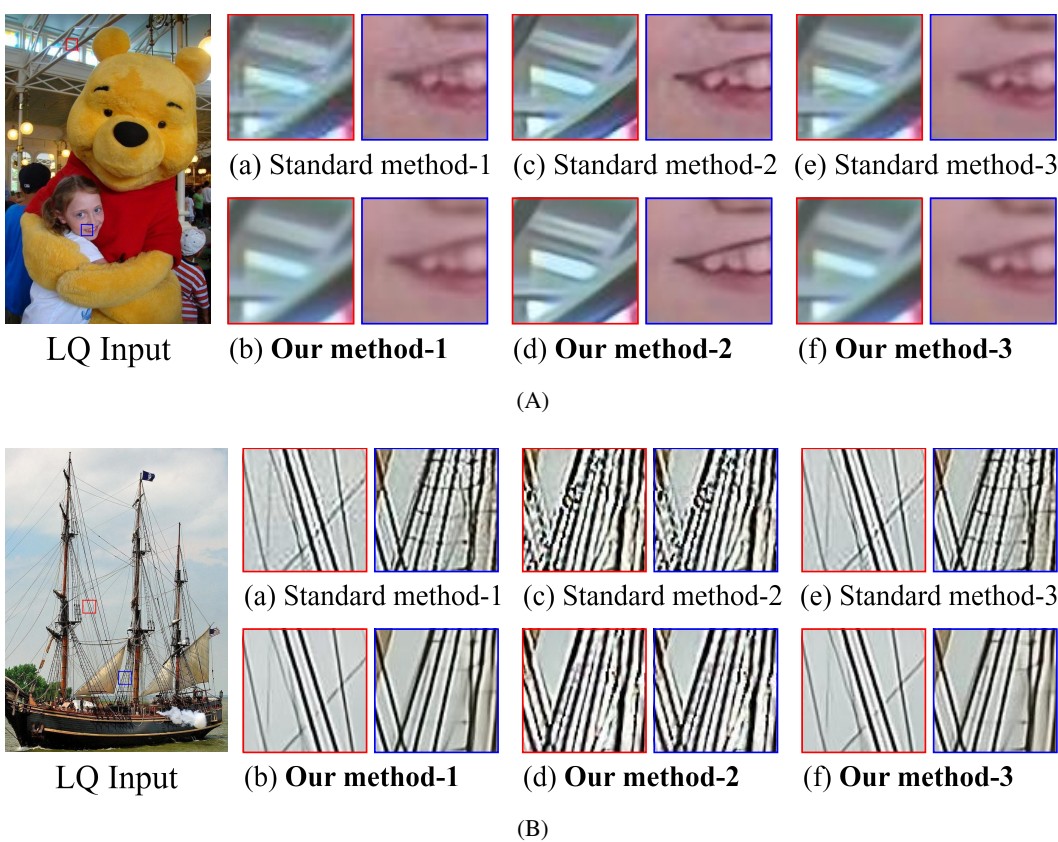

Figure 4: A visual demonstration of SwinIR's performance in restoring images with unknown degradation from real-world scenarios, trained using HQ-LQ pairs constructed from various data degradation methods. (a) resizing (downsampling), (b) first employing Stable Diffusion then resizing, (c) first using 4x bicubic downsampling then resizing, (d) first employing Stable Diffusion then using 4x bicubic downsampling and resizing, (e) first adding noise then resizing, (f) first employing Stable Diffusion then adding noise and resizing.

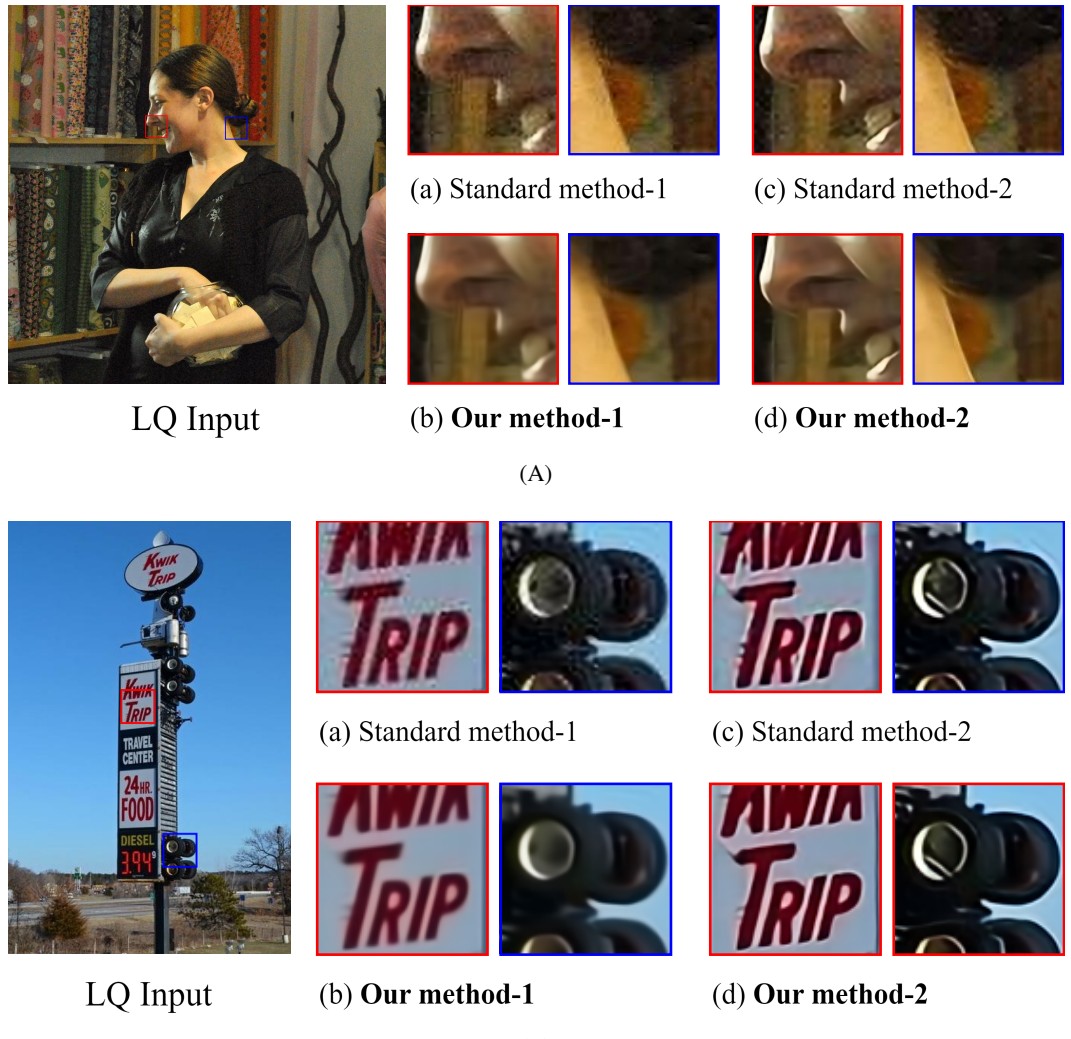

Figure 5: A visual demonstration of the MambaIR's performance in restoring images with unknown degradation from real-world scenarios, trained using HQ-LQ pairs constructed from various data degradation methods. (a) resizing (downsampling), (b) first employing Stable Diffusion then resizing, (c) first using 4x bicubic downsampling then resizing, (d) first employing Stable Diffusion then using 4x bicubic downsampling and resizing.

