# OpenReview forum: "Beyond Transformations: Augmenting Anything for Image Super-Resolution via Diffusion Model"
_ICLR.cc/2025/Conference — ICLR 2025 Conference Withdrawn Submission_

### Official Review · Reviewer_NzWW · 2024-10-22

**Soundness:** 3
**Presentation:** 2
**Contribution:** 2
**Rating:** 5
**Confidence:** 4

**Summary:**

This paper proposes a novel data augmentation method and an innovative data degradation technique in super-resolution (SR) modeling. The approach is straightforward, yet in some experiments, it achieves surprisingly significant improvements

**Strengths:**

1.The paper employs Stable Diffusion for data augmentation and degradation.

2.The approach is straightforward, yet in some experiments, it achieves surprisingly significant improvements.

**Weaknesses:**

**1. Lack of Insights**

The techniques proposed are quite similar, but offering no interesting insights (as there are a lot of works in image modification). As a result, the paper feels more like a technical report of a single method: how to use Stable Diffusion to modify images by changing the prompts for SR. The proposed Data Augmentation method alters image content through prompt modifications, while the Data Degradation method modifies image quality without altering the content. Overall, the methods seem straightforward and but offer no additional insights.

**2. Experimental Setup and Comparison for DA Method**

Typically, super-resolution methods are trained on DIV2K and evaluated on benchmark datasets like Set5, Set14, BSD100, Urban100, Manga109, and General100. For super-resolution in image synthesis, experiments are usually conducted on datasets such as ImageNet or FFHQ. However, this paper conducts experiments on animeSR, Manga109, and iCartoonFace, which is an unusual choice. This unconventional setup makes it difficult to assess the contributions compared to prior work. The paper only compares two approaches: Transformation-based and Diffusion-based methods, omitting other relevant baselines. It would be more beneficial for the authors to first train the baseline without data augmentation and then apply different data augmentation methods to demonstrate performance improvements, as done in [1][2].

**3. Lack of Quantitative Comparisons for DD Method**

Currently, Section 4.3 is dedicated to evaluating the Data Degradation method (DD), but it only provides qualitative comparison images (Figure 3). It would be more informative to include quantitative comparisons with relevant prior works, such as the well-known studies [3][4].

---
[1]. Reflash Dropout in Image Super-Resolution

[2]. Rethinking Data Augmentation for Image Super-resolution: A Comprehensive Analysis and a New Strategy

[3]. Real-esrgan: Training Real-world Blind Super-resolution with Pure Synthetic Aata

[4]. Designing a Practical Degradation Model for Deep Blind Image Super-Resolution

**Questions:**

I appreciate concise and effective work, and the approach presented in this paper is quite straightforward. However, there are still some issues that need to be addressed, which is why I am giving a slightly lower rating. The biggest concern is the comparison method discussed in the weaknesses section. (Currently, the paper only provides quantitative comparisons for DA and qualitative comparisons for DD. I recommend including both quantitative and qualitative comparisons for both DA and DD. Additionally, more comparative methods should be considered.)

Besides the issues mentioned in the weaknesses, there are some questions closely related to the paper's core ideas that, if addressed, would greatly enhance its persuasiveness:

**Q1.** Can DA still be considered data augmentation when it alters the image content? Wouldn't this be equivalent to introducing entirely new images?

**Q2.** Is DD truly capable of not altering image content at all? In Figure 2 (B), it’s quite clear that the shape of the tower tip has changed.

**Q3.** Why does DA lead to performance improvement?

I believe it's because DA modifies the image content, effectively introducing more training images. For instance, in traditional image super-resolution tasks, results on DIV2K alone aren't as good as when training on both DIV2K and Flickr2K. Table 3 in the paper seems to support this—adding more images continues to improve performance.

**Q4.** Is the comparison in Table 2 fair?

If the improvement is due to the introduction of more image content, experiments should ensure that the number of augmented images is consistent. Currently, there's a tenfold difference in the number of images, while the total number of iterations remains the same.

**Q5.** Is the reason for testing on a manga-style dataset because SD is particularly good at handling such style images?

**Q6.** Although the method appears straightforward, it seems somewhat challenging in practice. This is because it requires tuning both the strength value and the prompt. Prompts are notoriously unstable and can be very sensitive to specific words or phrases. For example, "add noise to image" may perform quite differently from "noisy image."

**Q7.** How is the DA prompt determined? What kind of new content should be generated for the best results? Is the process automatic or fixed across all datasets?

**Q8.** Should DA and DD be applied during each iteration, or should all images be pre-processed before training? How much additional time would this add compared to regular training?

---

### Official Review · Reviewer_Vxq4 · 2024-10-30

**Soundness:** 3
**Presentation:** 2
**Contribution:** 3
**Rating:** 3
**Confidence:** 4

**Summary:**

The paper employs Stable Diffusion to perform controlled data augmentation by introducing semantic variations to the image content without compromising the original quality. In addition, the paper extends the application of Stable Diffusion to simulate controlled data degradation, helping to make SR models more robust against real-world image variations and corruptions, thereby reducing restoration artifacts effectively.

**Strengths:**

The paper focuses on a dataset-enhancement application in stable diffusion for improving SR performance. The motivation is reasonable, and the perspective is novel.

**Weaknesses:**

1. The description of the proposed methods omits several important details, making it difficult to fully assess the approach's effectiveness.
A) The paper does not clearly explain what modification strength is and how it controls the SD model's output.
B) The fine-tuning process of SD (mentioned in line 211) lacks sufficient detail.
C) The authors should explain parameter alpha in Figure 2(A).
D) The authors should add a brief description of how the input is fed into the network in Figure 2(A) and (B), specifying whether and how noise is added in the latent space and any parameters used.
E) The negative prompt ("poor details") in Figure 2(B) shares attributes with the positive prompt, which is confusing, as one would expect the negative prompt to specify opposite attributes. The authors should clarify their rationale for this design.
F) Since SD is a text-to-image model that typically generates high-quality images, it is unclear how it can simulate real-world degradation. The authors should briefly explain how SD is adapted for this purpose, possibly with references to relevant literature.
G) To validate the proposed degradation method in Sec. 4.3, the authors should compare it with a well-established degradation pipeline, RealESRGAN [1]. The blind SR performance of models trained with different degradation methods could be assessed, using RealSR [2] as the test dataset and metrics such as PSNR and LPIPS for evaluation.

[1] Wang, Xintao, et al. "Real-esrgan: Training real-world blind super-resolution with pure synthetic data." Proceedings of the IEEE/CVF international conference on computer vision. 2021.
[2] Cai, Jianrui, et al. "Toward real-world single image super-resolution: A new benchmark and a new model." Proceedings of the IEEE/CVF international conference on computer vision. 2019.

2. Some experimental results lack sufficient analysis. In Table 1, while fidelity metrics (PSNR and SSIM) show significant improvement, perception-based metrics (LPIPS and DISTS) show little improvement. The authors should provide a brief analysis explaining why perception-based metrics do not improve as much.

**Questions:**

A) The authors should define modification strength and give the relevant reference, explain its impact on the output (e.g., does a higher value cause more drastic changes?), and clarify how appropriate values were determined for different tasks.

B) The authors should briefly explain which components were fine-tuned (VAE encoder, UNet, or VAE decoder) and the hyperparameters used. Additionally, since the SD model has been fine-tuned to the target data distribution, why not leverage its image generation capabilities to directly create diverse target images for training?

C) The authors should explain parameter alpha in Figure 2(A).

D) The authors should add a brief description of how the input is fed into the network in Figure 2(A) and (B), specifying whether and how noise is added in the latent space and any parameters used.

E) The negative prompt ("poor details") in Figure 2(B) shares attributes with the positive prompt, which is confusing, as one would expect the negative prompt to specify opposite attributes. The authors should clarify their rationale for this design.

F) Since SD is a text-to-image model that typically generates high-quality images, it is unclear how it can simulate real-world degradation. The authors should briefly explain how SD is adapted for this purpose, possibly with references to relevant literature.

G) To validate the proposed degradation method in Sec. 4.3, the authors should compare it with a well-established degradation pipeline, RealESRGAN [1]. The blind SR performance of models trained with different degradation methods could be assessed, using RealSR [2] as the test dataset and metrics such as PSNR and LPIPS for evaluation.

H) In Table 1, while fidelity metrics (PSNR and SSIM) show significant improvement, perception-based metrics (LPIPS and DISTS) show little improvement. The authors should provide a brief analysis explaining why perception-based metrics do not improve as much.

[1] Wang, Xintao, et al. "Real-esrgan: Training real-world blind super-resolution with pure synthetic data." Proceedings of the IEEE/CVF international conference on computer vision. 2021.
[2] Cai, Jianrui, et al. "Toward real-world single image super-resolution: A new benchmark and a new model." Proceedings of the IEEE/CVF international conference on computer vision. 2019.

---

### Official Review · Reviewer_tmQ3 · 2024-10-30

**Soundness:** 2
**Presentation:** 3
**Contribution:** 2
**Rating:** 3
**Confidence:** 5

**Summary:**

The paper presents a diffusion-based augmentation method for image super-resolution. While the traditional data augmentation methods are limited in data variance, the authors leverage the powerful pre-trained stable diffusion model to generate semantically controlled augmented samples. Moreover, the authors propose a similar way to yield degraded samples using pre-trained diffusion models. The extensive experiments on three super-resolution tasks have validated the effectiveness of the proposed augmentation methods.

**Strengths:**

1.	Leveraging pre-trained diffusion models to perform data augmentation is insightful in the field of image super-resolution. The authors target a rarely studied but crucial point for image super-resolution.
2.	The writing of the paper is clear. The experiments in the ablation study part are extensive and sufficient to prove the effectiveness of the proposed method.

**Weaknesses:**

1.	Although the ablation study part is extensive, there need more experiments in Sections 4.1, 4.2, and 4.3 to prove the superiority of diffusion-based augmentation.

$\quad$ a)	For Sections 4.1 and 4.2, my major concern is about the experiment settings. In both sections, setting (2) contains ten times as many images as setting (1), which leads to an unfair comparison between the two settings. As evidenced in [1], an image super-resolution model trained on a richer data distribution can achieve higher performance. Therefore, to better demonstrate the advantage of the proposed diffusion-based augmentation, the following settings may be more appropriate:

$\quad\quad$i.	Setting (1): randomly sample 1000 images from the FFHQ training set;

$\quad\quad$ii.	Setting (2): use 10000 images from the FFHQ training set;

$\quad\quad$iii.	Setting (3): apply diffusion-based augmentation to generate 10000 images based on the 1000 selected images.

$\quad$b)	For Section 4.3, my major concern is why the degradation pipeline introduced in [2,3] is not included as a comparison. This pipeline is proven to be effective in real application scenarios. Additionally, SwinIR offers a pre-trained model based on this degradation pipeline [4].

$\quad$c)	For Section 4.3, there are many existing paired and unpaired (only LR images) benchmarks for evaluation.

$\quad\quad$i.	Paired: RealSRV3[5], DrealSR[6].

$\quad\quad$ii.	Unpaired: RealSRSet[7], OST300[8].

It would be better to validate the effectiveness of the proposed method on more diverse real scenarios.

2.	Although diffusion model brings richer data distribution, some of the textures are unreal and anomalous. For example, texts in Figure 2(A) turn out to be corrupted after augmentation. The authors appear to lack the necessary methods to filter these unnatural samples.
3.	It would be better for authors to demonstrate more augmented samples in the paper to validate that the diffusion model is $\textbf{actually}$ following the prompt. For example, a noisy output with “add noise to the image”.
4.	From Table 1, 2, and 5, it seems that the proposed method is not beneficial for subjective metrics.
5.	There lacks references about diffusion-based data augmentation methods in other research fields: [9,10,11], and a highly related work [12].

[1] Li Y, Zhang K, Liang J, et al. Lsdir: A large scale dataset for image restoration[C]//Proceedings of the IEEE/CVF Conference on Computer Vision and Pattern Recognition. 2023: 1775-1787.

[2] Wang X, Xie L, Dong C, et al. Real-esrgan: Training real-world blind super-resolution with pure synthetic data[C]//Proceedings of the IEEE/CVF international conference on computer vision. 2021: 1905-1914.

[3] Zhang K, Liang J, Van Gool L, et al. Designing a practical degradation model for deep blind image super-resolution[C]//Proceedings of the IEEE/CVF International Conference on Computer Vision. 2021: 4791-4800.

[4] https://github.com/JingyunLiang/SwinIR/releases/tag/v0.0

[5] https://github.com/csjcai/RealSR

[6] https://github.com/xiezw5/Component-Divide-and-Conquer-for-Real-World-Image-Super-Resolution

[7] It is proposed by [3]

[8] https://github.com/xinntao/SFTGAN#ost-dataset

[9] Fu Y, Chen C, Qiao Y, et al. DreamDA: Generative Data Augmentation with Diffusion Models[J]. arXiv preprint arXiv:2403.12803, 2024.

[10] Feng C M, Yu K, Liu Y, et al. Diverse data augmentation with diffusions for effective test-time prompt tuning[C]//Proceedings of the IEEE/CVF International Conference on Computer Vision. 2023: 2704-2714.

[11] Trabucco B, Doherty K, Gurinas M, et al. Effective data augmentation with diffusion models[J]. arXiv preprint arXiv:2302.07944, 2023.

[12] Niu A, Zhang K, Tee J T J, et al. DifAugGAN: A Practical Diffusion-style Data Augmentation for GAN-based Single Image Super-resolution[J]. arXiv preprint arXiv:2311.18508, 2023.

**Questions:**

Please refer to weakness.

Would the proposed method be more effective for image super-resolution in AIGC?

---

### Author Response · Authors · 2024-11-15

We sincerely thank the reviewers and the Associate Chair for your insightful comments and suggestions. Your feedback has been invaluable, and we are committed to conducting further experiments to deepen our research. We appreciate the opportunity to improve our work based on your guidance.

---

### Note · Authors · 2024-11-15

**Comment:**

We sincerely thank the reviewers and the Associate Chair for your insightful comments and suggestions. Your feedback has been invaluable, and we are committed to conducting further experiments to deepen our research. We appreciate the opportunity to improve our work based on your guidance.

**Withdrawal Confirmation:**

I have read and agree with the venue's withdrawal policy on behalf of myself and my co-authors.